# Super-resolution microscopy of mitochondrial mRNAs

Stefan Stoldt[1,2,3,10], Frederike Maass[1,2,3,10], Michael Weber [4], Sven Dennerlein[5], Peter Ilgen [1,2,6], Jutta Gärtner[3,6,7,8], Aysenur Canfes [1,2], Sarah V. Schweighofer [1,6], Daniel C. Jans [1,2], Peter Rehling [3,5,6,8,9] & Stefan Jakobs [1,2,3,6,8] ✉

Mitochondria contain their own DNA (mtDNA) and a dedicated gene expression machinery. As the mitochondrial dimensions are close to the diffraction limit of classical light microscopy, the spatial distribution of mitochondrial proteins and in particular of mitochondrial mRNAs remains underexplored. Here, we establish single-molecule fluorescence in situ hybridization (smFISH) combined with STED and MINFLUX super-resolution microscopy (nanoscopy) to visualize individual mitochondrial mRNA molecules and associated proteins. STED nanoscopy reveals the spatial relationships between distinct mRNA species and proteins such as the RNA granule marker GRSF1, demonstrating adaptive changes in mRNA distribution and quantity in challenged mammalian cells and patient-derived cell lines. Notably, STED-smFISH shows the release of mRNAs during apoptosis, while MINFLUX reveals the folding of the mRNAs into variable shapes, as well as their spatial proximity to mitochondrial ribosomes. These protocols are transferable to various cell types and open new avenues for understanding mitochondrial gene regulation in health and disease.

Mitochondria are unique eukaryotic organelles, as they contain their own DNA (mtDNA). The mtDNA is a gene-dense circular DNA molecule of about 16.6 kb. A single mtDNA is compacted by proteins into a structure termed nucleoid, of which several hundred up to a few thousand exist in a cell[1–4].

The mitochondrial gene expression machinery that mediates the synthesis of only 13 proteins has a remarkable complexity and differs substantially both from the nuclear as well as from the bacterial machineries[5,6]. The mitochondrial DNA-directed RNA polymerase (POLRMT) initiates transcription on the light- and the heavy-strand,

producing near genome-length polycistronic transcripts that encompass all of the coding information of the respective strand[5,7]. Whereas 10 mRNAs (two are bicistronic), encoding for 12 subunits of the oxidative phosphorylation system (OXPHOS), two rRNAs, and 14 tRNAs are transcribed from the heavy strand, only one mRNA (*MT-ND6* mRNA), eight tRNAs, and several non-coding RNAs are transcribed from the light strand[8,9]. The polycistronic transcripts produced are processed within distinct foci, referred to as mitochondrial granules, to generate individual mRNAs, tRNAs, and rRNAs[10–12]. Biochemical analysis demonstrated low steady-state levels of the unprocessed

[1]Department of NanoBiophotonics, Max Planck Institute for Multidisciplinary Sciences, RG Mitochondrial Structure and Dynamics, Göttingen, Germany. [2]Clinic of Neurology, University Medical Center Göttingen, Göttingen, Germany. [3]Cluster of Excellence "Multiscale Bioimaging: from Molecular Machines to Networks of Excitable Cells" (MBExC), University of Göttingen, Göttingen, Germany. [4]Department of NanoBiophotonics, Max Planck Institute for Multidisciplinary Sciences, Göttingen, Germany. [5]Department of Cellular Biochemistry, University Medical Center Göttingen, Göttingen, Germany. [6]Fraunhofer Institute for Translational Medicine and Pharmacology, Translational Neuroinflammation and Automated Microscopy, Göttingen, Germany. [7]Department of Pediatrics and Adolescent Medicine, University Medical Center Göttingen, Göttingen, Germany. [8]German Center for Child and Adolescent Health (DZKJ), partner site Göttingen, Göttingen, Germany. [9]Max Planck Institute for Multidisciplinary Sciences, Göttingen, Germany. [10]These authors contributed equally: Stefan Stoldt, Frederike Maass. ✉e-mail: sjakobs@gwdg.de

transcripts, suggesting that processing of the transcripts occurs co-transcriptionally[13].

However, fundamental questions regarding the spatial distribution of the gene expression machinery in human mitochondria remain underexplored. In budding yeast, it has been shown that specific transcripts are preferentially translated at distinct sites within the mitochondrial inner membrane[14]. Yet, the sub-mitochondrial localizations and amounts of mitochondrial transcripts in individual mitochondria are unknown. Also, the spatial relation of specific mRNAs to nucleoids or to RNA granules are not known and the level of mRNA compaction within mitochondria has not been investigated. Although high-resolution profiling of mitochondrial translation and cryoelectron tomography suggested the existence of mitochondrial polysomes in human mitochondria[15,16], the existence of mitochondrial polysomes is still not conclusively proven. Arguably, to answer such questions, imaging is required. Although various fluorescent in situ hybridization (FISH) methods have been established for the study of cytoplasmic mRNAs, these approaches have been used very little to study mitochondrial transcripts[10,17–22]. A major challenge in imaging the localization of mitochondrial mRNAs is the small diameter of mitochondria, which necessitates the use of diffraction-unlimited super-resolution microscopy (nanoscopy) in most instances[23].

In this study, we tailored single molecule FISH (smFISH) of mitochondrial mRNAs for the use with STED and MINFLUX super-resolution microscopy. We report on several applications of smFISH in combination with STED microscopy, including the analysis of mRNA distribution and localization in cells with globally perturbed mitochondrial gene expression, in patient primary cells carrying a single t-RNA mutation as well as in apoptotic cells. Furthermore, we show that the overall shape and the distribution of mitochondrial mRNAs and associated proteins can be analyzed in the single-digit nanometer range using dedicated smFISH probes with MINFLUX nanoscopy.

## Results and discussion
### STED microscopy of mitochondrial mRNAs with smFISH
To robustly visualize the submitochondrial distribution of single mitochondrial mRNAs (mRNAs), we adapted branched DNA (bDNA) smFISH labeling for STED microscopy[24–26]. In bDNA FISH, pairs of primary probes hybridize to two adjacent regions of 20 to 30 nucleotides of the target RNA (Fig. 1A). Only pairs of primary probes properly bound to the target RNA are effectively hybridized by a preamplifier, thereby ensuring high specificity. For each mRNA, a different number of probe pairs (3–22) was used, based on the transcript length and sequence (Table 1), thereby ensuring a high transcript detection probability[24]. To ensure an optimal signal-to-noise ratio, preamplifiers were hybridized with multiple amplifiers, which in turn were hybridized with multiple label probes, each coupled to STED compatible fluorophores, resulting in an assembled amplification "tree" (Fig. 1A).

Using this approach, we imaged three different mitochondrial mRNAs (*MT-ND1, MT-CYB, MT-CO3* mRNA) which are transcribed from the mtDNA heavy strand in U-2 OS cells (Fig. 1B). In the three-color STED images, the individual mRNAs could be visualized as clearly discernible structures (Fig. 1C). Most mRNAs were spatially separated from one another; however, instances of close proximity between different pairs or even triplets of mRNAs were also observed (Fig. 1D).

The assembled amplification "tree" used for labeling could potentially extend for several 100 nm, resulting in a large fluorescent spot. This would lead to an underestimation of the degree of mRNA compaction within the mitochondria. To test for this, we analyzed the average diameter (full width at half maximum) of the fluorescent spots in the STED images. We found an average diameter of about 85 nm, suggesting the collapse of the branched "trees". These data thereby also suggest a high degree of mRNA compaction (Fig. 1E).

Analysis of the pairwise minimum distances between the three mRNAs revealed similar distributions with a median of around 200 nm (Fig. 1F). The pairwise distances between different mRNA species were similar to the pairwise distances of the same mRNA species (Supplementary Fig. 1A). The similarity in mRNA distances, irrespective of gene proximity on the mtDNA (gene positions on mtDNA: *MT-ND1*: 3307-4262; *MT-CO3*: 9207-9990; *MT-CYB*: 14747-15887) and also between the same mRNA species (which necessarily originate from distinct primary transcripts), are fully in line with previous findings that mitochondrial mRNAs are immediately excised from the primary transcript[13,27,28].

To further confirm the specificity of the smFISH probes for imaging mitochondrial mRNAs, we targeted all mitochondrial mRNAs at once in wild-type (WT) U-2 OS control cells and in U-2 OS cells without mtDNA (Rho0 cells) and consequently without mRNAs. In WT cells, this led to a strong signal throughout the mitochondrial network, whereas in Rho0 cells we observed a complete loss of signal, confirming the absence of non-specific probe binding (Supplementary Fig. 1B). To validate the specificity further, we tested a probe for rat *Mt-co3* mRNA on human cells (Supplementary Fig. 1C). Full length human *MT-CO3* mRNA and rat *Mt-co3* mRNA have a sequence homology of 78%. Despite the high sequence similarity, we were unable to detect any signal in human cells with the anti-rat smFISH probe, further confirming the high specificity of the probes.

To investigate, if the established labeling and imaging protocols can be combined with the labeling of proteins and are transferable to other cell types, cells were labeled with probes targeting *MT-CO1* mRNA or *MT-CYB* mRNA, as well as with antibodies labeling dsDNA and proteins such as GRSF1 and TOM20, respectively (Supplementary Fig. 2). We employed the smFISH probes on several widely used cell types such as HEK-293, HeLa, human primary fibroblasts, as well as on rat hippocampal neurons and combined smFISH with immunolabeling. In all sample types we were able to label the mRNAs together with mitochondrial proteins and/or mtDNA and visualize their specific distributions using STED microscopy (Supplementary Fig. 2).

Together, a robust smFISH protocol for three-color STED super-resolution imaging of three different mitochondrial mRNAs or mRNAs in combination with immunolabeling of mitochondrial DNA and proteins was established.

### MINFLUX imaging of mitochondrial mRNAs
STED-smFISH facilitated the analysis of the distribution of mitochondrial mRNAs within the mitochondrial network. As the achievable resolution of 2D-STED nanoscopy in cellular samples is generally limited to around 30–40 nm[29], this approach is not suited to visualize the fold of a single mRNA.

MINFLUX nanoscopy enables the localization of fluorophores with single-digit nanometer precision in 3D by using a local excitation minimum for localization, making it the most photon-efficient super-resolution technique available[30]. The attainable nominal resolution is an order of magnitude higher than in STED microscopy, although at the cost of long recording times. In MINFLUX nanoscopy, stochastic switching is utilized. DNA-PAINT has proven to be particularly useful for this purpose and has been shown to enable the detection of multiple targets through sequential MINFLUX imaging[31]. Hence, we developed MINFLUX-smFISH probes with the preamplifier probe directly fused to a DNA-PAINT docking strand (Fig. 2A). This design of smFISH probes for DNA-PAINT MINFLUX allowed for the stochiometric labeling of the probe pairs and ensured a minimal size of the labels.

Using these probes, we sequentially imaged three mitochondrial mRNAs (*MT-ND1, MT-CO1, MT-CYB* mRNA) in segments of mitochondrial tubules (Fig. 2B). In the MINFLUX-smFISH data representation, the localization of a single fluorophore is rendered as one sphere of 10 nm. A cloud of spheres thereby represents the extent and structure of an mRNA. We observed, fully in line with the STED data, often separated individual mRNAs, but also clusters of two or three mRNAs

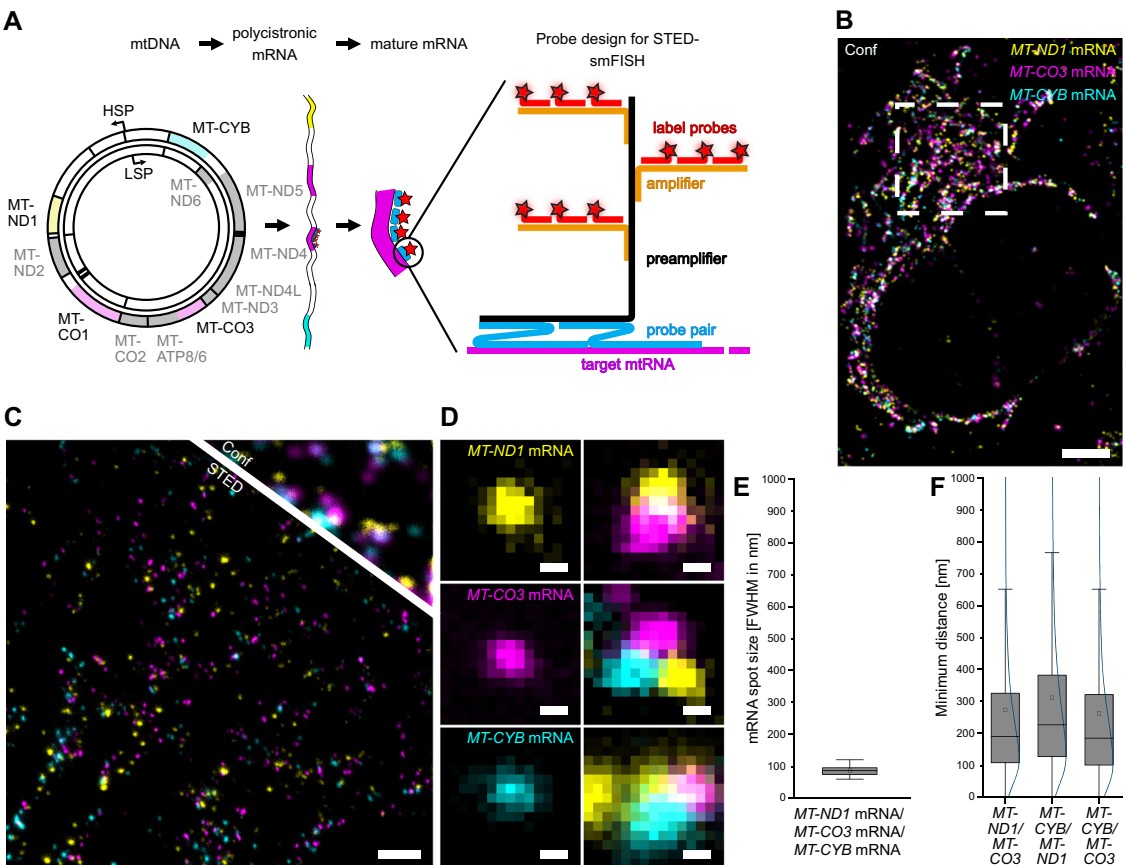

**Fig. 1 | STED-smFISH probe design and super-resolution imaging of mitochondrial mRNAs. A** Schematic representation of the STED-smFISH probe design and probe binding. From left to right: Both strands of the circular mtDNA are transcribed into polycistronic mRNAs, which are further processed to generate 11 mature mRNAs. The STED-smFISH probes consist of probe pairs (blue) that target a specific mRNA (magenta). Several probe pairs, depending on the length of the transcript, can bind simultaneously. Only when both probes of a pair are bound, the preamplifier (black) can bind. Multiple amplifier strands (orange) then bind to the preamplifier. The probe 'tree' is completed by the binding of the label probes (red), which are coupled to the fluorophore of choice (red stars). Specific probe pairs are available for all mRNAs (see Table 1). HSP: heavy strand promoter, LSP: light strand promoter. **B** Three-color confocal overview image of a U-2 OS cell labeled for the *MT-ND1* mRNA (yellow), *MT-CO3* mRNA (magenta), and *MT-CYB* mRNA (cyan). **C** Comparison between the STED and the confocal image at a higher magnification of the indicated area in (**B**). **D** Exemplary close-ups of spatially separated mRNAs (left) and mRNA clusters (right). **E** Full width half maximum (FWHM) of *MT-ND1*, *MT-CO3*, and *MT-CYB* mRNA spot sizes. N = 3, *n* = 171, mtRNA spots: 6674. **F** Quantification of the minimum pairwise distances between *MT-ND1*, *MT-CO3*, and *MT-CYB* mRNAs. N = 3, *n* = 70. Box: 25/75 percentile; whiskers: max/min without outliers; line: median; square: mean (**E**, **F**). Curved line: kernel density estimation representing the data point distribution (**F**). Source data are provided as a Source Data file. 'N' indicates biological replicates; 'n' indicates technical replicates. Scale bars: 5 μm (B), 1 μm (C), 50 nm (D).

(Fig. 2C). In the MINFLUX data, the experimentally determined median localization precision of the fluorophores was below 3 nm in all three dimensions in all data sets (Fig. 2D). As the fluorophores are close to the mRNA and distributed along its length, the distribution of the fluorophores reflects the shape of the mitochondrial mRNA. On the resolution scale of single molecule MINFLUX imaging, we found that the mRNAs adopted a variety of structures, from compact to elongated (Fig. 2E).

Since the RNA is lined by the fluorophores in MINFLUX-smFISH, we could use the closest fluorophores belonging to different mRNAs to determine the minimum distance between two different mRNAs. As expected, the average minimum distances of different mRNAs determined by MINFLUX-smFISH (median from 75 nm to 91 nm; Fig. 2F) was lower than in STED-smFISH data (median from 184 nm to 226 nm; Fig. 1F), as in the latter the centers of distinct fluorescence distributions were used as reference points. Nevertheless, no relevant differences in the pairwise minimum distances between the different mRNAs could be detected in the MINFLUX-smFISH images (Fig. 2F).

To compare the results obtained by MINFLUX nanoscopy with STED images, we flattened the 3D MINFLUX data into a virtual plane

and convolved it with a 40 nm FWHM gaussian function (Supplementary Fig. 3A), thereby emulating STED images based on MINFLUX data. The size of the fluorescent spots in the emulated STED data was on average 50 nm (Supplementary Fig. 3B), which is smaller than found in real STED images (85 nm; Fig. 1E) taken from the same samples. The measured minimum distances between mRNA species in the emulated data (median from 68 nm to 100 nm; Supplementary Fig. 3C) were very close to those found in the MINFLUX data (Fig. 2F) and lower than in the real STED data (Fig. 1F). The differences between the real STED data and the emulated STED data may be attributed to the size differences of the label probes; notably, however, the number of mRNA positions that can be recorded in a reasonable amount of time is by orders of magnitude higher in STED imaging than in MINFLUX microscopy, making the real STED data statistically more robust than the MINFLUX-based emulated STED data.

Labeling of all mitochondrial mRNAs and subsequent MINFLUX imaging showed that the mRNAs are localized in clusters along the mitochondrial tubules (Supplementary Fig. 3D). This was particularly evident when comparing their distribution with that of the mitochondrial large ribosomal subunit protein bL12m (MRPL12) labeled

**Table 1 | Design of probe pairs used in this study for smFISH**

| Mitochondrial mRNA | Length in nt | Number of simultaneously bound probe pairs | Transcript coverage by probe pairs (nt) | Transcript coverage by probe pairs (%) |
|---|---|---|---|---|
| MT-ND1 | 954 | 7 | 322 | 34 |
| MT-ND2 | 1041 | 9 | 417 | 40 |
| MT-ND3 | 345 | 5 | 243 | 70 |
| MT-ND4L | 294 | 5 | 201 | 68 |
| MT-ND4 | 1377 | 7 | 343 | 25 |
| MT-ND5 | 1809 | 5 | 451 | 25 |
| MT-ND6 | 522 | 3 | 137 | 26 |
| MT-CO1 | 1539 | 15 | 696 | 45 |
| MT-CO2 | 681 | 4 | 188 | 28 |
| MT-CO3 | 783 | 5 | 221 | 28 |
| Mt-co3 (rat) | 814 | 16 | 683 | 84 |
| MT-ATP8 | 204 | 3 | 143 | 70 |
| MT-ATP6 | 678 | 5 | 218 | 32 |
| MT-CYB | 1140 | 18 | 767 | 67 |
| MT-CYB (rat) | 1157 | 22 | 954 | 82 |

Note that MT-ATP6/8 and MT-ND4L/4 are biscistronic mRNAs.

with a primary labeled antibody (Supplementary Fig. 3E). In contrast to mitochondrial mRNAs, bL12m exhibited a homogeneous distribution throughout the mitochondrion. Occasionally, multiple bL12m foci were observed in proximity to mitochondrial mRNAs, potentially indicating the presence of polysomes (Supplementary Fig. 3F).

### STED-smFISH: perturbed processing of the polycistronic transcripts

To determine whether STED-smFISH enables the visualization of changes in the number and distribution of specific mRNAs in cultured U-2 OS cells, we first downregulated the catalytic subunit of mtRNAse P (Protein Only RNase P Catalytic Subunit; PRORP or MRPP3), which cleaves off the tRNAs that intersperse the protein-coding mRNAs at their 5′-end[32–34] (Supplementary Fig. 4A). Previous studies analyzing the average amounts of mRNAs in cells after downregulation of PRORP reported an increase of mitochondrial mRNAs[27,35]. Fully in line with this, STED-smFISH showed that upon PRORP knockdown the ratio of mRNA clusters to mtDNA clusters was increased 2.1-fold (MT-ND1 mRNA), 1.4-fold (MT-CO1 mRNA), 1.9-fold (MT-ATP6 mRNA), and 1.3-fold (MT-CYB mRNA) compared to the WT (Fig. 3A, B).

The guanine-rich RNA sequence binding factor 1 (GRSF1) is a signature protein of mRNA granules[10,11,36,37]. We found that the knockdown of PRORP resulted in an aggregation of GRSF1 into large spots and an accumulation of mRNAs at these sites (Fig. 3A). As these aggregates were surrounded by numerous mtDNAs, the structures remind of previously described RNA granules[10,11]. Such GRSF1 extended, nucleoid surrounded RNA granules were not apparent in unperturbed U-2 OS cells when imaged with STED microcopy, indicating a complex and dynamic behavior of these structures (Fig. 3A).

### STED-smFISH: perturbed mitochondrial transcription

To investigate whether also impaired mitochondrial transcription influences the localization of mitochondrial mRNAs and GRSF1, we inhibited the mitochondrial DNA-dependent RNA polymerase (POLRMT) in U-2 OS cells with the specific inhibitor IMT1 (inhibitor of mitochondrial transcription 1)[38]. Treatment with 10 μM IMT1 for 20 h resulted in lower mRNA and mtDNA levels as well as a homogeneous distribution of GRSF1 in the mitochondrial matrix (Supplementary Fig. 4B). Specifically, the number of MT-CO1 mRNAs and mitochondrial nucleoids was reduced fivefold and twofold,

respectively (Supplementary Fig. 4C), fully in line with previous biochemical studies on cell populations, which showed that inhibition of POLRMT causes a strong decrease in mitochondrial mRNAs and a gradual depletion of mtDNA[38].

### STED-smFISH in apoptotic cells

Next, we explored STED-smFISH to visualize mitochondrial mRNAs in apoptotic cells. During apoptosis, mitochondria round up and rupture, as the effector proteins BAX and BAK oligomerize and form apoptotic pores, leading to the release of mitochondrial contents, ultimately resulting in cell death. While the release of nucleoids into the cytoplasm during apoptosis has been shown using super-resolution microscopy[39], little is known about the release of mRNAs, and in particular imaging-based approaches to study this process are lacking. Therefore, we investigated whether STED-smFISH allows to visualize the release of mRNAs during apoptosis. To induce apoptosis, U-2 OS cells were treated for 16 h with 20 μM Actinomycin D (ActD) and 10 μM ABT-737. After chemical fixation, we labeled for MT-CO1 mRNA, BAX, and the mitochondrial outer membrane protein TOM22. As expected, mitochondria in apoptotic cells fragmented and BAX accumulated on the mitochondrial outer membrane (Fig. 3C). Using STED-smFISH, we visualized MT-CO1 mRNAs in the apoptotic cells. The recordings demonstrate that MT-CO1 mRNAs were indeed partially released during apoptosis and were often found near the BAX pores (Fig. 3C). Concomitantly, the number of MT-CO1 mRNAs confined by the mitochondrial outer membrane decreased by 65% (Fig. 3D).

Taken together, these examples of cultured cells treated with inhibitors of mtDNA replication and mitochondrial transcription, or inducers of apoptosis demonstrate that STED-smFISH is able to reveal changes in the number and localization of mitochondrial mRNAs along with mitochondrial proteins.

### STED-smFISH demonstrates changed mRNA distributions in patient cells

Finally, to investigate the potential of STED-smFISH for the study and characterization of cells of clinical relevance, we analyzed mRNA abundances in primary fibroblasts derived from a patient harboring a single point mutation (m.14709 T > C) in the mitochondrially encoded tRNA-Glu (MT-TE)[40,41]. For this, we labeled MT-CO1 mRNA together with mtDNA and GRSF1 (Fig. 3E). We found a small, but significant decrease in MT-CO1 mRNAs in the diseased mitochondria compared to mitochondria in healthy control cells (Fig. 3F). This reduction is also reflected in the ratio of the number of mRNA clusters to the number of mtDNA clusters (Supplementary Fig. 4D). Moreover, the number of nucleoids per mitochondrial area is increased in the diseased cells (Supplementary Fig. 4E), while the number of GRSF1 clusters per mitochondrial area is reduced in these cells (Supplementary Fig. 4F). Thus, the STED-smFISH data suggests that a single point mutation in a mitochondrial tRNA leads to defects in mRNA processing and thus to a lower abundance of mRNA and other factors involved in mitochondrial gene expression. STED-smFISH enables the visualization of such subtle differences.

### Concluding remarks and limitations of the study

In summary, we developed robust FISH protocols for multi-color imaging of single mitochondrial mRNAs using STED and MINFLUX super-resolution microscopy. The data indicate a tight folding of the mRNAs and reveal the distribution of mRNAs in mitochondria. We can identify subtle differences in the distribution of mRNAs in challenged cultured mammalian cells as well as in cells from patients with mitochondrial disorders. The method paves the way to study RNA metabolism and mitochondrial transcription with nanometer spatial resolution in cellulo.

This study is primarily a method report, demonstrating the potential of the method to visualize localizations and shapes of mitochondrial mRNAs. Detailed biological conclusions, such as the

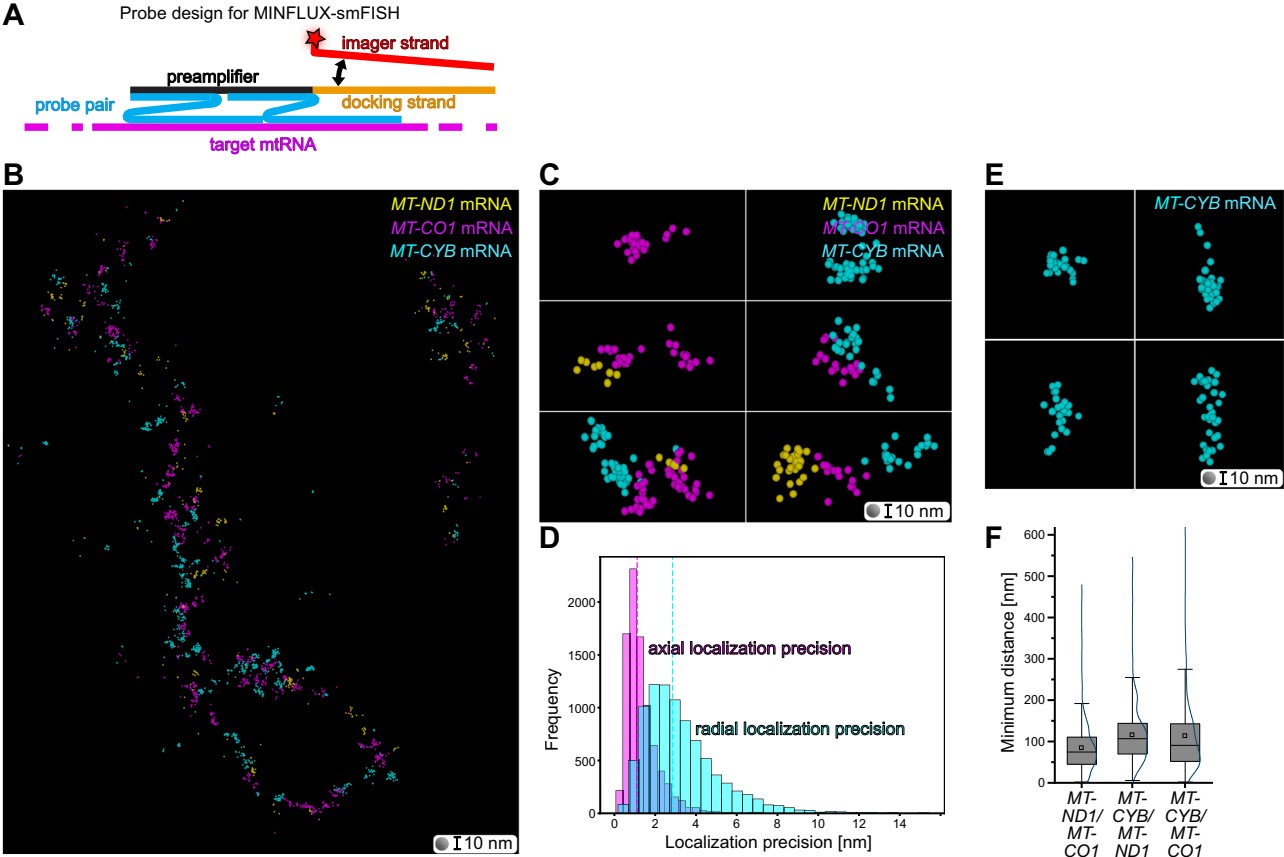

**Fig. 2 | Design of MINFLUX-smFISH probes and 3D nanometer-scale mapping of mitochondrial mRNAs. A** Schematic representation of the MINFLUX-smFISH probe design. MINFLUX-smFISH probes consist of probe pairs (blue) targeting a specific mRNA (magenta). Both probes of a pair must bind to the mRNA in order for the preamplifier (black) to bind. The preamplifier for MINFLUX-smFISH is contiguous with a docking strand for DNA-PAINT (orange). The complementary imager strand (red), which is coupled to a single fluorophore (red star), transiently binds to the docking strand. **B** 3D rendition of MINFLUX localizations of sequentially acquired *MT-ND1* mRNA (yellow), *MT-CO1* mRNA (magenta), and *MT-CYB* mRNAs (cyan) data in a mitochondrial segment (rendered with sphere diameters of 10 nm). **C** Close-ups of several sites from (**B**). **D** Histogram of the distribution of combined localization precisions of MINFLUX data. Dashed lines: median localization precisions. Magenta: axial localization precision (median = 1.1 nm). Cyan: radial localization precision (median = 2.8 nm). **E** Example close-ups from (**B**) showing different shapes of *MT-CYB* mRNA reflected by the fluorophore positions. **F** Quantification of the minimum pairwise distances between *MT-ND1*, *MT-CO3*, and *MT-CYB* mRNAs. N = 2, *n* = 4. Box: 25/75 percentile; whiskers: max/min without outliers; line: median; square: mean; curved line: kernel density estimation representing the data point distribution (**F**). Source data are provided as a Source Data file. 'N' indicates biological replicates; 'n' indicates technical replicates. Sphere diameter = 10 nm (**B**, **C**, **E**).

processing state of nearby mRNAs, will require additional experimental approaches.

A potential concern is the size of the probes used to label mRNAs for STED imaging, as a fully expanded, assembled amplification "tree" could potentially span the diameter of a mitochondrion. However, the STED data strongly suggest a collapse of the label probes on the mRNAs, minimizing this concern. Comparison of emulated STED images (based on MINFLUX data obtained using smaller probes) and real STED data (using assembled amplification "trees") suggested a slight increased size (by ~35 nm) of mRNAs in the STED data. This shows that the impact of the amplification "tree" on the size measurements was minimal (Fig. 1E, Supplementary Fig. 3B).

3D MINFLUX-smFISH enables the analysis of mRNA distributions and shapes in 3D with nanometer resolution. However, in MINFLUX nanoscopy, fluorophore positions are determined one at a time, resulting in long imaging times and comparatively small imaged volumes. As a result, the statistical basis of the recorded MINFLUX data is rather weak. In contrast, STED microscopy, in the implementation used here, provides a large amount of 2D data in a comparatively short time. Therefore, even small differences in the RNA distribution between different samples can be detected reliably and with high statistical significance. The choice of imaging method must therefore be determined by the requirements for resolution and statistical robustness of the data.

## Methods
### Animals

All procedures with live animals were conducted in the animal facility at the Max Planck Institute for Multidisciplinary Sciences. According to the German Animal Welfare Law, sacrificing an animal is not regarded as an experiment on animals. All requirements of section 4 TierSchG together with sections 2 Satz 2, Anlage 1, Abschnitt 2, and Anlage 2 TierSchVersV were implemented.

The facility is conducted under all aspects of animal welfare. The facility is headed by a veterinarian with special education in laboratory animal science as well as gene technology and molecular genetics.

Only professionally educated animal technicians are in charge of animal husbandry and care. The facility is registered according to section 11 Abs. 1 TierSchG (Tierschutzgesetz der Bundesrepublik Deutschland, Animal Welfare Law of the Federal Republic of Germany) as documented by 33.23-42508-066-§11, dated 16 November 2023 (Erlaubnis, zum Halten von Wirbeltieren zur Versuchszwecken, permission to keep vertebrates for experimental purposes) by the Niedersächsisches Landesamt für Verbraucherschutz und

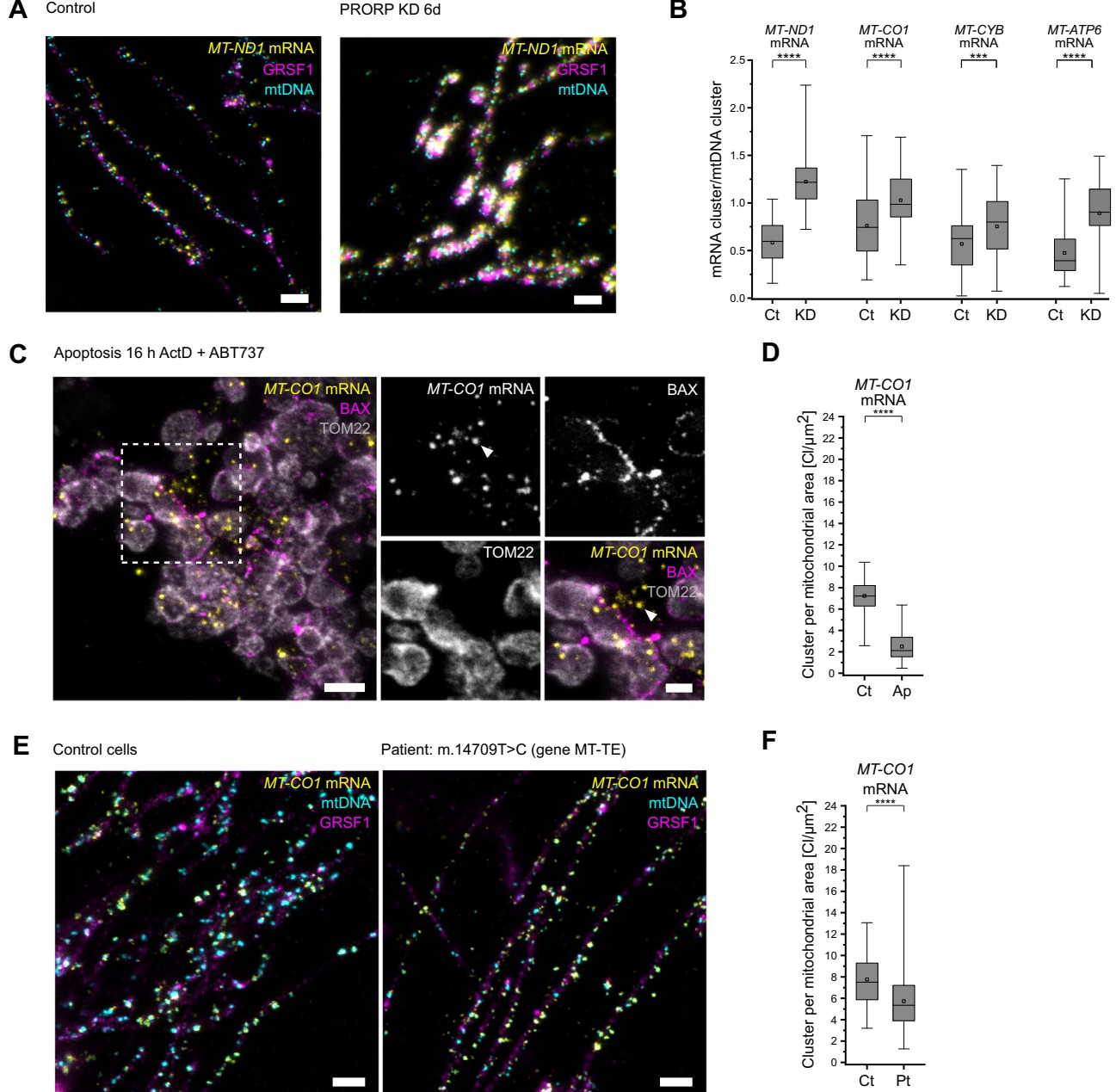

**Fig. 3 | STED-smFISH to visualize nanoscale changes in mitochondrial mtRNA localizations. A** STED-smFISH of U-2 OS cells: control cells treated with scrambled siRNA and PRORP knockdown (KD) cells, labeled for *MT-ND1* mRNA (yellow), GRSF1 (magenta), and mtDNA (cyan). **B** Quantification of the *MT-ND1*, *MT-CO1*, *MT-CYB*, and *MT-ATP6* mRNA cluster to mtDNA cluster (nucleoid) ratio in control cells (Ct) and in PRORP KD cells (KD). N = 3, n(*MT-ND1*, Ct) = 60, n(*MT-ND1*, KD) = 62, n(*MT-CO1*, Ct) = 73, n(*MT-CO1*, KD) = 73, n(*MT-CYB*, Ct) = 74, n(*MT-CYB*, KD) = 73, n(*MT-ATP6*, Ct) = 73, n(*MT-ATP6*, KD) = 71; ****: *p*-value <0.0001, ***: *p*-value <0.001 . **C** STED image (overview and magnifications as indicated) of an apoptotic U-2 OS cell (treated with ActD, ABT-737, and Q-VD-OPh for 16 h). Cells were labeled for BAX (magenta), TOM22 (gray), and *MT-CO1* mRNA (yellow) and show *MT-CO1* mRNA signals outside of the confines of the mitochondrial outer membrane (arrowhead). **D** Quantification of *MT-CO1* mRNA clusters per mitochondrial area in untreated control (Ct) cells and apoptotic (Ap) cells. N = 3, n(Ct) = 74, n(Ap) = 74; *p* value <0.0001. **E** STED-smFISH of *MT-CO1* mRNA (yellow) together with immuno-labeling for GRSF1 (magenta) and mtDNA (cyan) in cultured human fibroblasts, derived from a control (left) and a patient sample (right) carrying the m.14709 T > C mutation in gene MT-TE encoding the tRNA-Glu. **F** Quantification of *MT-CO1* mRNA clusters per mitochondrial area in control fibroblasts (Ct) and fibroblasts from the patient (Pt). N = 3, n(Ct) = 74, n(Pt) = 75; *p* value <0.0001. Box: 25/75 percentile; whiskers: max/min without outliers; line: median; square: mean; Statistical analysis between two groups (**B**, **D**, **F**) was performed using an unpaired, two-sided Student's t test. 'N' indicates biological replicates; 'n' indicates technical replicates. Source data are provided as a Source Data file. Scale bars: 1 μm (**A**, **C** overview, **E**) and 500 nm (**C** close-ups).

Lebensmittelsicherheit (Lower Saxony State Office for Consumer Protection and Food Safety). According to the Animal Welfare Law of the Federal Republic of Germany (TierSchG) and the Regulation about animals used in experiments, dated 20 December 2022 (TierSchVersV) an animal welfare officer (specialized veterinarian in laboratory animal science) and an animal welfare committee for the institute was established.

Adult Wistar rats were obtained from Charles River Laboratories or Janvier. All animals were maintained on a 12-h light/12-h dark cycle with ad libitum access to food and water until use.

## Cell culture

Four different kinds of human cells were used for this study: Human osteosarcoma cells (U-2 OS, ECACC92022711), human cervical cancer cells (HeLa CCL-2), human embryonic kidney 293 (HEK-293, CVCL_U421) cells, human WT fibroblasts, and human WT fibroblasts from a patient with the substitution mutation m.14709 T > C (gene MT-TE) in the mitochondrial tRNA coding sequence. HeLa, HEK-293, and human fibroblasts were cultivated at 37 °C and 5% $CO_2$ in DMEM, containing 4.5 g/L Glucose and GlutaMAX™ additive (Thermo Fisher #10566016), supplemented with 1 mM sodium pyruvate (Sigma Aldrich #S8636) and 10% (v/v) (FBS) (Bio&SELL #FBS.S0615). U-2 OS cells were cultivated at 37 °C and 5% $CO_2$ in McCoy's medium (Thermo Fisher Scientific #16600082) supplemented with 1 mM sodium pyruvate and 10% (v/v) FBS.

## Hippocampal neuron culture

WT *Rattus norvegicus* (strain Wistar) pups of unknown sex were sacrificed at p0 and the hippocampi from both hemispheres were extracted in cooled HBSS buffer (Thermo Fisher Scientific #14175095). The hippocampi were digested in 0.25% trypsin (Thermo Fisher Scientific #15090046) for 18 min at 37 °C. Afterwards, the digestion was blocked by adding DMEM (Thermo Fisher Scientific #31966047) and after centrifugation and washing with HBSS buffer, the hippocampi were mechanically dissected by pipetting up and down with a serological pipette in Neurobasal medium (Thermo Fisher Scientific #12349015). The cell suspension was plated onto sterile cover slips (1.5H 18 mm cover slips, Marienfeld #0117580) coated with 100 µg/mL Poly-L-ornithine (Merck Millipore #A-004-C). After 1 h, the medium was changed to Neurobasal medium supplemented with 1× GlutaMAX (Thermo Fisher #35050061), 1× Pen/Strep (Merck Millipore #P0781), 1× B27 supplement (Thermo Fisher #11530536) and cells were grown for 19 days.

## KD with siRNAs

U-2 OS cells were seeded on cover slips and let attach overnight. They were then treated with gene specific siRNA pools or negative control siRNA (siTOOLs Biotech, PRORP: Gene ID: 9692, bL12m: 6182). The siRNAs were pre-diluted in $ddH_2O$ and then to a concentration of 48 nM in OptiMEM (ThermoFisher #11058021). The solution was mixed with Lipofectamine RNAiMAX (Invitrogen #13778150) (diluted in OptiMEM, according to the manufacturer's protocol) to a concentration of 24 nM, vortexed for 30 sec, and added to the cells in culture medium to a final concentration of 6 nM. The cells were incubated for 6 days in case of the PRORP KD. For the bL12m KD, cells were transfected again after 3 days and incubated for 8 days in total.

## Rho0 cells

U-2 OS POLRMT knockout cells were used as Rho0 cells. The POLRMT KO cell line was created using CIRSPR/Cas9. In short, the Benchling (Benchling, Inc.) tool was used to design guide RNAs (oligos) and their reverse complements for the target gene (POLRMT). The guide RNA and its reverse complement were hybridized in the thermocycler and cloned into the pX458 pSpCas9(BB)−2A-GFP (Addgene, plasmid #48138) plasmid. Cells were transfected with the cloned plasmid, containing the respective guide RNA and Cas9 nuclease. JetPrime (Polyplus-trasfection #101000046) reagent was used for the transfection according to manufacturer´s instructions. Cells expressing Cas9-EGFP were single-cell sorted in 96-well plates using a benchtop Sony SH800S Cell Sorter (Sony Biotechnology Inc.) three days after transfection. After clonal cell expansion, the single-cell clones were confirmed by immunoblotting, PCR of the target gene, sub-cloning with TOPO® (Zero Blunt™ TOPO™ PCR Cloning Kit, Invitrogen™) vector, and subsequent Sanger sequencing.

## Apoptosis induction

U-2 OS cells were treated for 16 h with 20 µM Actinomycin D (ActD) (Merck Millipore #A1410), 10 µM ABT-737 (APExBIO #A8193) to induce apoptosis and 10 µM pan-caspase inhibitor (Q-VD-OPh) (APExBIO #A1901) to prevent the detachment of the cells from the coverslip.

## Western blotting

U-2 OS cells were harvested and lysed in lysis buffer (50 mM TRIS [VWR #103156X], 1% sodium dodecyl sulfate (SDS) [Serva #20765], 0.1 mM dithiothreitol [AppliChem #A1101,0025], 4 mM $MgCl_2$ [Thermo Fisher #AM9530G], 1× cOmplete protease inhibitor [Merck #11873580001], 4 mM phenylmethylsulfonyl fluoride [Thermo Fisher #36978], 0.01 U/µL benzonase nuclease [Merck #E1014]) for 2 h at 4 °C. The protein concentration was determined by utilizing the QubitTM Protein Broad Range (BR) Assay Kit (Thermo Fisher #Q33211). The samples (20 µg) were mixed with 4× Laemmli buffer (0.25 M TRIS [VWR #103156X], 0.28 M SDS [Serva #20765], 40% glycerol [VWR #24385.295], 20% 2-mercaptoethanol [Sigma Aldrich #M6250-100Ml], 6 mM bromophenol blue [AppliChem #A3640,0025]) and denatured for 10 min at 95 °C. The samples were loaded on the NuPAGE™ 4 to 12% Bis-TRIS gel (Thermo Fisher #WG1401BX10) and run with 1× NuPAGE™ 2-Morpholinoethanesulfonic acid monohydrate (MES) SDS running buffer (Thermo Fisher #NP0002) at 130 V for 75 min. Afterwards, the gel was washed in 20% ethanol for 10 min and transferred on nitrocellulose membrane with the iBlotTM transfer stack (Thermo Fisher #IB301001) at 20 V for 1 min, 23 V for 4 min, and 25 V for 2 min. The membrane was washed in ddH20 and 1× TRIS buffered saline (TBS) (Thermo Fisher #J60764.K2), followed by 2 h blocking in 5% milk in TBS with Tween (TBST) (Thermo Fisher #J77500.K2) at room temperature. After 15 min washing in 1× TBST, the membrane was incubated with the primary antibodies diluted in 2% milk in TBST overnight at 4 °C. After three times washing in TBST for 10 min, the membrane was incubated with secondary antibodies diluted in 2% milk in TBST for 1 h at room temperature. Lastly, the membrane was washed again in TBST, incubated for 1 min in Immobilon Forte Western HRP Substrate (Merck #WBLUF0100), and imaged with an Amersham Imager 600 (GE Healthcare). Antibodies and antibody-dilutions used: anti-β-actin (Sigma Aldrich #A5441, 1:2000), anti-MRPP3 (Abcam #ab185942, 1:1000), goat-anti-rabbit HRP (Jackson ImmunoResearch #111-035-144, 1:10000), goat-anti-mouse HRP (Jackson ImmunoResearch #115-035-062, 1:10000).

## Design of probe pairs used in this study for smFISH

Probe pairs were designed according to the *View RNA™ Cell Plus Assay* by ThermoFisher (Table 1). The number of simultaneously bound probe pairs per mRNA ranged from 3 to 18 depending to the length and sequence of the transcript. Accordingly, the transcript coverage by the probe pairs ranges from 25 to 70%.

## Coupling of dyes to label probes for STED-smFISH

To generate custom label probes with dyes suitable for STED microscopy, 5'-amino functionalized label probe DNA (5' Amino Modifier C6) of Type 1, Type 4, and Type 6 (*ViewRNA Cell Plus Assay*) were purchased from ThermoFisher (custom order). Labeling of the functionalized label probe DNA with a fluorophore was performed as reported earlier for the labeling of 3'-amino functionalized oligonucleotides[42]. In brief, the amino-modified DNA (10 nmol) was dissolved in carbonate buffer (100 mM, pH 9.0) and the solution of the respective succinimidyl ester activated dyes (Table 2) in DMF (10 mM) was added. The reaction mixture was incubated at 37 °C in an incubator shaker for 3 h in the dark. Subsequently, labeled DNA was purified by denaturing PAGE with subsequent gel extraction and ethanol precipitation.

## Labeling of samples with smFISH for STED microscopy

RNA FISH labeling was performed using the ViewRNA ISH Cell Plus Assay by ThermoFisher with custom labeled label probes

(https://www.thermofisher.com/de/de/home/life-science/cell-analysis/cellular-imaging/in-situ-hybridization-ish/rna-fish/viewrna-cell-assays.html) according to the supplier's protocol with adaptations. Cells were seeded on 1.5H glass cover slips and pre-fixed with 2% formaldehyde (Merck Millipore #No. 30525-89-4) in 1× PBS for 10 min at 37 °C. After this pre-fixation step, an additional fixation and permeabilization was carried out using the *ViewRNA Cell Plus Fixation/Permeabilization Solution* for 20 min at 37 °C. The cells were then washed 5 times in 1× PBS with *ViewRNA Cell Plus RNase Inhibitor* and then incubated in *ViewRNA Cell Plus Probe Solution* containing the probe pairs (Table 1) for 2 h at 40 °C. After incubation with the probe pairs, the cells were washed 5 times with *ViewRNA Cell Plus RNA Wash Buffer Solution* followed by incubation in *ViewRNA Cell Plus PreAmplifier Solution* containing the preamplifier, which binds to the bound probe pair, for 1 h at 40 °C. After 5 times washing in *ViewRNA Cell Plus RNA Wash Buffer Solution*, the cells were incubated in *ViewRNA Cell Plus Working Amplifier Solution*, which contains the amplifier strands binding to the preamplifier, for 1 h at 40 °C. Lastly, after 5 times washing in View*RNA Wash Buffer Solution* the cells were incubated for 1 h at 40 °C with *ViewRNA Cell Plus Working Label Probe Mix Solution* with the added custom labeled label probes at 1.5 pmol/μL, 0.5 pmol/μL, and 1 pmol/μL for probe Type 1, 4, and 6, respectively. After the final 5 washing steps, the samples were either mounted on glass slides (ThermoFisher #11562203) embedded in Mowiol (2.4 g PVA in 6 g glycerol, 6 mL ddH$_2$O, 12 mL 200 mM Tris Buffer pH 8.5 plus 2.5% (w/v) DABCO) or immunofluorescence labeling was performed.

### Immunofluorescence labeling
Immunofluorescence labeling was performed after smFISH labeling. The samples were blocked for 5 min with 5% BSA (AppliChem #A1391) in 1× PBS. The cells were then incubated with primary antibodies diluted in BSA for 1 h at room temperature (Table 3). After 6 times washing in 1× PBS, cells were incubated with secondary antibodies in 5% BSA for 1 h at room temperature. Afterwards, cells were washed 6 times in 1× PBS, dipped in ddH$_2$O and mounted (as described above).

### Custom labeling of primary antibody for DNA-PAINT MINFLUX
For ribosome immunofluorescence labeling for MINFLUX together with MINFLUX-smFISH, the rb-anti-MRPL12 (anti-bL12m) antibody was azide-activated with the GlyCLICK Azide activation kit (GENOVIS #L1-AZ1-025) and coupled to the P3 DNA handle[43].

### Design of preamplifiers for MINFLUX-smFISH
For MINFLUX-smFISH, the custom preamplifiers of the Type 1, Type 4, and Type 6 (*ViewRNA Cell Plus Assay*) with the DNA-PAINT docking strands P3, P1, and P2, respectively[43], were purchased from Thermo Fisher Scientific.

### Labeling of samples for MINFLUX nanoscopy
MINFLUX-smFISH labeling was carried out in the same way as for STED-smFISH until the addition of the preamplifiers (see above). For MINFLUX-smFISH the custom MINFLUX-smFISH preamplifiers were used (see above). After 5 times washing in *ViewRNA Cell Plus RNA Wash Buffer Solution*, samples were kept in PBS until DNA-PAINT MINFLUX imaging or immunofluorescence labeling with the custom labeled primary antibody was performed as described above. Finally, the samples were kept in PBS until DNA-PAINT MINFLUX imaging.

### Confocal/ STED imaging
Confocal imaging was performed using a TCS SP8 (Leica, Wetzlar, Germany) with a HC PL APO CS2 63x/1.40 Oil objective (Leica) (software: Leica Application Suite X 3.5.7.23225) or an Abberior Expert Line Quad-Scanning STED microscope (Abberior Instruments) with a UPlanSApo 100x/1.40 Oil objective (Olympus) (software: Imspector16.1.-win64-AIFpgaV3). For confocal imaging with the TCS SP8 (Supplementary Fig. 3E), Alexa Fluor 488 was excited at 496 nm and STAR RED at 633 nm. For the confocal and STED imaging with the Abberior Expert Line STED microscope (all other image data sets), Alexa Fluor 488 was excited at 485 nm, Alexa Fluor 594 at 561 nm and STAR RED at 640 nm. STED was applied at 595 nm (Alexa Fluor 488) or 775 nm (Alexa Fluor 594, STAR RED), respectively. For all STED image acquisitions, a pixel size of 20 nm was used. Recording parameters were used as described previously[44]. Except for contrast stretching, no further image processing was carried out for representation of the data.

### DNA-PAINT MINFLUX imaging
Sample mounting and 3D MINFLUX measurements were performed as described previously[31]. In brief, an Abberior MINFLUX microscope (Abberior Instruments) was used for imaging. The Imspector software (16.3.11636-m2205-win64-MINFLUX, Abberior Instruments) and the MINFLUX sequence template "seqDefault3d_w2101" were used for the 3D MINFLUX acquisitions. MINFLUX imaging was performed with a

### Table 2 | List of fluorophores used for labeling

| Fluorophore reagent | Company + Cat. No. |
|---|---|
| Alexa Fluor™ 488 NHS ester (succinimidyl ester) | Thermo Fisher Scientific #A20000 |
| Alexa Fluor™ 594 NHS ester (succinimidyl ester) | Thermo Fisher Scientific #A20004 |
| Abberior STAR RED NHS carbonate | Abberior #STRED-0002-1MG |

### Table 3 | List of antibodies used for immunofluorescence labeling

| Antibody | Company + Cat. No., dilution |
|---|---|
| Primary ms-anti-BAX | Thermo Fisher #MA5-13994, 1:50 |
| Primary ms-anti-dsDNA | Abcam #ab27156, 1:2000 |
| Primary rb-anti-GRSF1 | Sigma-Aldrich #HPA036985, 1:400 |
| Primary rb-anti-Mitofusin 2 | Cell Signaling Technologies, #9482, 1:100 |
| Primary rb-anti-TOM20 | Abcam #ab186734, 1:200 |
| Primary rb-anti-TOM22 | Sigma-Aldrich #HPA003037, 1:400 |
| Primary rb-anti-bL12m | Proteintech #14795-1-AP, 1:400 |
| Primary ms-anti-ATP Synthase Subunit beta | Abcam #ab5432, 1:200 |
| Secondary goat-anti-rabbit Alexa488 | Molecular Probes #A11034, 1:200 |
| Secondary goat-anti-mouse Oregon Green | Molecular Probes #O6380, 1:200 |
| Secondary goat-anti-rabbit Alexa594 | Invitrogen #A11037, 1:200 |
| Secondary sheep-anti-mouse StarRed | Jackson ImmunoResearch #515005062, custom labeled StarRed NHS carbonate Aberrior #STRED-0002-1MG, 1:200 |

640 nm excitation laser at a laser power of 71 μW (16%) in the first iteration in the sample, and a pinhole size of 0.57 AU (referenced to the emission maximum of Atto 655 at 680 nm; 0.67 in the Imspector software with a reference wavelength of 574 nm) was used. Before mounting them on stage, the cells were labeled with probe pair and preamplifier + docking strand and were incubated for 5 min with fiducials for beam stabilization (150 nm gold beads (BBI Solutions #SKU EM.GC150/4)) and washed five times with PBS. Imager strands P1, P2, and P3 (all conjugated to Atto655) were diluted separately in imaging buffer (Massive Photonics) to a concentration of 5 nM and sequentially applied to the sample on the microscope stage. Between the acquisitions, the sample was washed five times and the next imaging strand was applied. During imaging, beam monitoring on about five gold beads was activated.

## Statistical analysis

All experiments were performed at least in triplicates. For each experiment, parts of the mitochondrial network of at least 20 cells (in total for each case ≥60 cells) were imaged.

Statistical analyses were performed using GraphPad Prism 9. The data was checked for normality by visual inspection of Q-Q-Plots. For normally distributed data sets, an unpaired two-tailed t test was employed. The significance level ****, ***, **, and * correspond to a *p* value of <0.0001, <0.001, <0.01, and <0.05, respectively. In the cases where kernel density estimation of the data points was performed, this was conducted using a bandwidth selected based on Scott's rule.

## STED data analysis

The STED data analysis was performed using as custom written MATLAB script (MATLAB R2018b). In short, the STED images were smoothed with a 25 nm FWHM Gaussian filter, and background removal was performed by subtracting a 210 nm FWHM Gaussian filtered image from the smoothed image. mRNA localizations were defined at positions of local maxima in areas that were at least 100 nm away from the image edges and exceeded at least 3 detected counts after background subtraction (Supplementary Fig. 5). To determine the mitochondrial area from the (background subtracted) images, a circular area with a radius of 6 pixels (120 nm) around the local maxima was used. Areas smaller than 200 pixels were discarded. The resulting binary mask was convolved with a 5×5 pixel square. The resulting images were thresholded (pixel-intensity 12.5) and thus binarized again. Then, holes in the mitochondrial mask smaller than 200 pixels were filled. The detected mRNA, protein or mtDNA cluster positions inside the mitochondria were counted and divided by the mitochondrial mask area to obtain cluster densities. For distance calculations, the minimum pairwise distances of the detected mRNAs were calculated. The boxplots were created with OriginPro 2020 (64-bit) (OriginLab).

## MINFLUX data analysis

The MINFLUX data processing and analysis was performed using as custom written Python script. For each ROI the sequentially acquired MINFLUX data of each channel was aligned based on the gold bead positions from the beam monitoring. Localization precisions were calculated as previously described[31]. In brief, the individual localizations from each binding event with more than four localizations were combined. The stated localization precision in x, y ($s_r$) and z ($s_z$) refers to the median of the localization precisions of the combined localizations. Next, we determined the axis with the worst precision for each dataset and discarded all combined localizations with a precision below two times its mean. To filter out background localizations, the combined localizations of all datasets of one ROI were concatenated and DBSCAN with epsilon of 120 nm and minimum 8 points was performed on the data. The clustered datapoints were used for visualization and further analysis. The minimal Euclidean distance of each

(filtered) combined localization to a combined localization of a different dataset of the same ROI was calculated. The minimum distance boxplots were created with OriginPro 2020 (64-bit) (OriginLab) and the combined localizations were plotted using Blender 3.4.1 with a sphere size (diameter) corresponding to 10 nm.

## Reporting summary

Further information on research design is available in the Nature Portfolio Reporting Summary linked to this article.

## Data availability

All primary data are available at Zenodo https://doi.org/10.5281/zenodo.15631472. Source data are provided with this paper.

## Code availability

STED data analysis code is available at Zenodo https://doi.org/10.5281/zenodo.15650078 and the MINFLUX data processing and analysis code is available at Zenodo https://doi.org/10.5281/zenodo.15650178.

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

## Acknowledgements

We thank the facility for Synthetic Chemistry of the MPI-NAT (head: Dr. Vladimir Belov) for coupling of the label probes, especially Jan Seikowski. We thank Ellen Rothermel, Nicole Molitor, and Anne Folmeg for technical support. We thank the Animal Facility of the MPI-NAT for the animal caretaking. This work was supported by the European Research Council (ERCAdG No. 835102) (SJ), (ERCAdG No. 101054637) (PR), by the graduate program RNApp (A2, SJ), by the German Center for Neurodegenerative Diseases, DZNE (JG), by the DFG-funded FOR2848 (P04, PR and SJ), TRR 274 (B02, JG), SFB 1286 (A05, SJ; A06, PR), by the German Center for Child and Adolescent Health, DZKJ (SJ, JG, PR) (BMBF 01GL2402A) and by the DFG under Germany's Excellence Strategy - EXC 2067/1- 390729940 (SJ, JG, PR).

## Author contributions

PR and SJ designed research. SJ supervised the study. SS, FM, SD, AC, SVS, and DCJ performed research. SS, FM, MW, PI, and DCJ analyzed data. JG and PR provided materials. SS, FM, SVS, DCJ, and SJ wrote the manuscript. All authors edited and approved the final manuscript.

## Funding

## Competing interests

The authors declare no competing interests.
