## [Transparent Peer Review file · Nature Communications]

Super-resolution microscopy of mitochondrial mRNAs

Corresponding Author: Professor Stefan Jakobs

Version 0:

Reviewer comments:

Reviewer #1

(Remarks to the Author)

The authors have shifted the focus of their work on the technique, however, this is not sufficient to warrant publication as the key experiments suggested by all the reviewers have not been addressed.

Reviewer #2

(Remarks to the Author)

The reviewers have revised the focus of the manuscript towards the technical and methodologic aspects of their RNA super resolution data, and this revised focus as well as the other changes are satisfactory.

Reviewer #3

(Remarks to the Author)

I appreciate the revised direction and structure of the manuscript, which emphasizes the technical aspects and lays a strong foundation for future biological investigations into mitochondrial RNA structure and dynamics. I am pleased with most of the changes the authors have made in response to my previous comments. I support the publication of this study following minor revisions to address the following points.

1. Page 4, Lines 104-106: Branched DNA “tree” swelling issue.

The reported FWHM of 85 nm indicates that the labeled structure exceeded the STED resolution of 40 nm, suggesting that mRNA is compacted. However, the true size of mRNA may be smaller than 85 nm. To address this, I recommend convolving the MINFLUX images in Fig. 2 with the STED PSF, as the authors already performed, measuring the FWHM, and comparing it with the STED FWHM of the branched structures. Since MINFLUX labeling uses shorter preamplifier sequences and omits amplifier sequences, the FWHM derived from MINFLUX labeling could serve as a control to verify the effects of branched labeling.

2. Page 4, Line 112: Gene proximity clarification

The text references gene pairs in proximity or originating from distinct primary transcripts, but it does not specify which genes are being discussed. To support the argument, please specify the names of the gene pairs.

3. Page 4, Lines 112-114: Branched labeling effects on cluster counts

In Figure 3 and on pages 6-8, the cluster counts derived from STED-smFISH are the primary metric used to compare wild-type and treated conditions (e.g. siRNA, knockdown, drug treatment, apoptosis, or patient-derived cells). However, the potential impact of branched labeling on cluster counts is not addressed. For instance, expanded branches from neighboring mRNA in close proximity could appear as a single cluster, under-sampling the cluster counts. I recommend performing MINFLUX imaging for one of the conditions in Figure 3. Then convolve the MINFLUX images with the STED PSF and compare the resulting cluster counts to those in STED-smFISH to assess the influence of branched labeling on the results.

4. Page 8, Lines 234-246: Wild-type vs. patient cell images

While the images of wild-type and patient-derived cells in Figure 3E appear notably different, the analysis results in Figure 3F show only slight differences. For instance, the mRNA and DNA clusters in the patient-derived cells appear far less

abundant than those in wild-type, yet the box plot indicates only a small decrease in the average values, albeit with the large error bar. This discrepancy raises concerns about whether the images are representative of the data. If the current images are not representative, consider replacing the examples in Figure 3E with more typical or consistent images to better reflect the analysis results.

Version 1:

Reviewer comments:

Reviewer #3

(Remarks to the Author)

The authors have satisfactorily addressed my concerns. I support the publication of the revised manuscript.

Reviewer #1 (Remarks to the Author):

The authors have shifted the focus of their work on the technique, however, this is not sufficient to warrant publication as the key experiments suggested by all the reviewers have not been addressed.

We revised the text to clearly emphasize that our manuscript is primarily focused on methodology. Specifically, we added a new paragraph discussing the study's limitations and acknowledged that some of the biological conclusions presented are preliminary and not deeply explored.

Concretely it reads (page 9):

Limitations of this study

This study is primarily a method report, demonstrating the potential of the method to visualize nanoscale localizations and shapes of mitochondrial mRNAs. Detailed biological conclusions, such as the processing state of nearby mRNAs, will require additional experimental approaches.

A potential concern is the size of the probes used to label mRNAs for STED imaging, as a fully expanded, assembled amplification "tree" could potentially span the diameter of a mitochondrion. However, the STED data strongly suggest a collapse of the label probes on the mRNAs, minimizing this concern. Comparison of emulated STED images (based on MINFLUX data obtained on smaller probes) and real STED data (using assembled amplification "trees") suggested a slight increased apparent size (by ~35 nm) of mRNAs in the STED data. This shows that the impact of the amplification "tree" on the size measurements was minimal (Fig. 1E, Suppl. Fig. 3B).

3D MINFLUX-smFISH enables the analysis of mRNA distributions and shapes in 3D with nanometer resolution. However, in MINFLUX nanoscopy, fluorophore positions are determined one at a time, resulting in long imaging times and comparatively small imaged volumes. As a result, the statistical basis of the recorded MINFLUX data is rather weak. In contrast, STED microscopy, in the implementation used here, provides a large amount of 2D data in a comparatively short time. Therefore, even small differences in the RNA distribution between different samples can be detected reliably and with high statistical significance. The choice of imaging method must therefore be determined by the requirements for resolution and statistical robustness of the data.

Reviewer #2 (Remarks to the Author):

The reviewers have revised the focus of the manuscript towards the technical and methodologic aspects of their RNA super resolution data, and this revised focus as well as the other changes are satisfactory.

We thank the reviewer for his/her positive view on the manuscript and the constructive and helpful comments.

Reviewer #3 (Remarks to the Author):

I appreciate the revised direction and structure of the manuscript, which emphasizes the technical aspects and lays a strong foundation for future biological investigations into mitochondrial RNA structure and dynamics. I am pleased with most of the changes the authors have made in response to my previous comments. I support the publication of this study following minor revisions to address the following points.

We thank this reviewer for his/her helpful comments.

1. Page 4, Lines 104-106: Branched DNA “tree” swelling issue.

The reported FWHM of 85 nm indicates that the labeled structure exceeded the STED resolution of 40 nm, suggesting that mRNA is compacted. However, the true size of mRNA may be smaller than 85 nm. To address this, I recommend convolving the MINFLUX images in Fig. 2 with the STED PSF, as the authors already performed, measuring the FWHM, and comparing it with the STED FWHM of the branched structures. Since MINFLUX labeling uses shorter preamplifier sequences and omits amplifier sequences, the FWHM derived from MINFLUX labeling could serve as a control to verify the effects of branched labeling.

We thank the reviewer for raising this important point. We followed exactly his/her suggestion and emulated STED data based on the MINFLUX data shown in Fig. 2. As suggested, we determined the resulting FWHM of *MT-ND1*, *MT-CO1* and *MT-CYB* mRNA spot sizes, and, in addition, quantified the minimum pairwise distances between various pairs of mRNAs in the emulated data and compared it to real STED data (see new Suppl. Fig. 3A-C). As predicted by this reviewer, we found that the branched labelling affects the apparent size of the mRNAs. Concretely, the FWHM of the RNA spots in real STED data was on average 85 nm, and in the emulated STED images based on MINFLUX data it was on average 50 nm. We attribute this difference primarily to the branched labelling.

It reads (page 6):

To compare the results obtained by MINFLUX nanoscopy with STED images, we flattened the 3D MINFLUX data into a virtual plane and convolved it with a 40 nm FWHM gaussian function (Suppl. Fig. 3A), thereby emulating STED images based on MINFLUX data. The size of the fluorescent spots in the emulated STED data was on average 50 nm (Suppl. Fig. 3B), which is smaller than found in real STED images (85 nm; Fig. 1E) taken from the same samples. The measured minimum distances between mRNA species in the emulated data (median from 68 nm to 100 nm; Suppl. Fig. 3C) were very close to those found in the MINFLUX data (Fig. 2F) and lower than in the real STED data (Fig. 1F). The differences between the real STED data and the emulated STED data may be attributed to the size differences of the label probes; (...).

And (page 9):

A potential concern is the size of the probes used to label mRNAs for STED imaging, as a fully expanded, assembled amplification “tree” could potentially span the diameter of a mitochondrion. However, the STED data strongly suggest a collapse of the label probes on the mRNAs, minimizing this concern. Comparison of emulated STED images (based on MINFLUX data obtained on smaller probes) and real STED data (using assembled amplification “trees”) suggested a slight increased apparent size (by ~35 nm) of mRNAs in the STED data. This shows that the impact of the amplification “tree” on the size measurements was minimal (Fig. 1E, Suppl. Fig. 3B).

2. Page 4, Line 112: Gene proximity clarification

The text references gene pairs in proximity or originating from distinct primary transcripts, but it does not specify which genes are being discussed. To support the argument, please specify the names of the gene pairs.

Done. It now reads:

Analysis of the pairwise minimum distances between the three mRNAs revealed similar distributions with a median of around 200 nm (Fig. 1F). The pairwise distances between different mRNA species were similar to the pairwise distances of the same mRNA species (Suppl. Fig. 1A). The similarity in mRNA distances, irrespective of gene proximity on the mtDNA (gene positions on mtDNA: MT-ND1: 3307-4262; MT-CO3: 9207-9990; MT-CYB: 14747-15887) and also between the same mRNA species (which necessarily originate from distinct primary transcripts), are fully in line with previous findings that mitochondrial mRNAs are immediately excised from the primary transcript.

3. Page 4, Lines 112-114: Branched labeling effects on cluster counts

In Figure 3 and on pages 6-8, the cluster counts derived from STED-smFISH are the primary metric used to compare wild-type and treated conditions (e.g. siRNA, knockdown, drug treatment, apoptosis, or patient-derived cells). However, the potential impact of branched labeling on cluster counts is not addressed. For instance, expanded branches from neighboring mRNA in close proximity could appear as a single cluster, under-sampling the cluster counts. I recommend performing MINFLUX imaging for one of the conditions in Figure 3. Then convolve the MINFLUX images with the STED PSF and compare the resulting cluster counts to those in STED-smFISH to assess the influence of branched labeling on the results.

We thank the reviewer for this insightful comment. As discussed above and shown in the newly added Suppl. Fig. 3A-C, comparing emulated STED data derived from MINFLUX measurements with real STED images reveals a small but clear influence (~35 nm) of the assembly "tree" on the FWHM of mRNA spots.

Assessing the effect of the assembly "tree" on cluster counts would indeed be informative. As the reviewer rightly noted, this would require STED and MINFLUX images of the same condition, followed by convolution of the MINFLUX data with a STED point spread function.

In fact, an important finding of this manuscript is the substantial variability in mRNA distributions both between cells and among individual mitochondria within a single cell. To ensure statistically meaningful comparisons between conditions, extensive imaging datasets are required. While such datasets are feasible to obtain using 2D STED microscopy - which is why we feature it multiple times throughout the manuscript - they are currently not attainable with MINFLUX microscopy due to its lengthy acquisition times, making statistically sound analyses impractical. For reference, the STED image in Fig. 1c was acquired in ~ 4 minutes, whereas the MINFLUX image in Fig. 2B covering a considerably smaller area - required approximately eight hours of acquisition time.

Consequently, performing the suggested comparison using MINFLUX data is not feasible within a reasonable timeframe. Our conclusion, now clearly stated in two locations in the revised manuscript, is that while 3D MINFLUX-smFISH offers unmatched spatial resolution for analyzing the 3D organization and morphology of mRNAs at the nanometer scale, STED microscopy provides statistically more robust data due to its superior throughput.

It reads (page 6):

(...) ; notably, however, the number of mRNA positions that can be recorded in a reasonable amount of time is by orders of magnitude higher in STED imaging than in MINFLUX microscopy, making the real STED data statistically more robust than the MINFLUX-based emulated STED data.

And (page 9):

3D MINFLUX-smFISH enables the analysis of mRNA distributions and shapes in 3D with nanometer resolution. However, in MINFLUX nanoscopy, fluorophore positions are determined one at a time, resulting in long imaging times and comparatively small imaged volumes. As a result, the statistical basis of the recorded MINFLUX data is rather weak. In contrast, STED microscopy, in the implementation used here, provides a large amount of 2D data in a comparatively short time. Therefore, even small differences in the RNA distribution between different samples can be detected reliably and with high statistical significance. The choice of imaging method must therefore be determined by the requirements for resolution and statistical robustness of the data.

4. Page 8, Lines 234-246: Wild-type vs. patient cell images

While the images of wild-type and patient-derived cells in Figure 3E appear notably different, the analysis results in Figure 3F show only slight differences. For instance, the mRNA and DNA clusters in the patient-derived cells appear far less abundant than those in wild-type, yet the box plot indicates only a small decrease in the average values, albeit with the large error bar. This discrepancy raises concerns about whether the images are representative of the data. If the current images are not representative, consider replacing the examples in Figure 3E with more typical or consistent images to better reflect the analysis results.

We fully agree with the reviewer's comment - the shown image of the patient cell line was poorly chosen. It has been replaced with a more representative image.